# Incremental Schema Discovery at Scale for RDF Data

Redouane Bouhamoum, Zoubida Kedad, and Stéphane Lopes

DAVID lab. - University of Versailles Saint-Quentin-en-Yvelines
Versailles, France
`firstName.lastName@uvsq.fr`

**Abstract.** The lack of a descriptive schema for an RDF dataset has motivated several research works addressing the problem of automatic schema discovery. The goal of these approaches is to generate a structural schema of a given RDF dataset from its instances. However, as new instances are added, the generated schema may become inconsistent with the dataset.

In this paper, we propose an incremental schema discovery approach for massive RDF datasets. It is based on a scalable and incremental density-based clustering algorithm which propagates the changes occurring in the dataset into the clusters corresponding to the classes of the schema. Our approach is implemented using big data technology to scale up schema discovery while providing a high quality clustering result. We present some experiments which demonstrate the efficiency of our proposal on both synthetic and real datasets.

**Keywords:** Incremental Schema Discovery · RDF Data · Big Data · Clustering.

## 1 Introduction

The Web of data represents a huge information space consisting of an increasing number of interlinked datasets described using the Resource Description Framework (RDF)[1]. One important feature of such datasets is that they contain both the data and the schema describing the data. However, these schema-related declarations are not mandatory, and are not always provided. As a consequence, the schema may be incomplete or missing.

The lack of schema offers a high flexibility while creating interlinked datasets, but can also limit their use. Indeed, it is not obvious to query or explore a dataset without any knowledge on its resources, classes or properties. The exploitation of an RDF dataset would be easier with a schema describing the data.

We have proposed in previous works a schema discovery approach suitable for very large datasets, which relies on a scalable density-based clustering algorithm [3]. It enables fast density-based clustering on large datasets and provides

---

[1] RDF: https://www.w3.org/RDF/

a good quality schema. However, RDF datasets are subject to frequent evolutions over time, and new instances may be inserted. For example, between version 3.5 and version 3.9 of DBpedia[2], the number of triples having the class *Person* as their object has been multiplied by 45 [14]. Due to such evolution, the ability to perform incremental updates on the schema has emerged as a new challenge.

In this work, we introduce an incremental schema discovery approach for large RDF datasets. Our contribution is an incremental density-based clustering algorithm for building and updating the clusters that represent the classes of the schema. Our algorithm incrementally updates the classes describing an RDF dataset in order to keep the schema consistent with the evolution of the data and ensures that the result is the same as if the clustering algorithm has been executed on the whole dataset in one go. In addition, the incremental clustering process is parallelized to be efficient on large datasets. The source code of the implementation of our algorithm, based on the distributed processing framework Apache Spark[18] is available online [3].

The rest of the paper is organized as follows. Section 2 presents the problem addressed in this paper and provides some preliminary notations. The general idea of our approach is introduced in section 3. Section 4 presents our data distribution principle. Section 5 describes the computation of the neighborhood of the newly inserted entities. Section 6 presents the generation of the new schema. Experimental results are presented in section 7, and section 8 discusses the related works. Finally, a conclusion is provided in section 9.

## 2   Problem Statement

An RDF *dataset* is a set of RDF(S)/OWL triples $\mathcal{D} \subseteq (\mathcal{R} \cup \mathcal{B}) \times \mathcal{P} \times (\mathcal{R} \cup \mathcal{B} \cup \mathcal{L})$, where $\mathcal{R}$, $\mathcal{B}$, $\mathcal{P}$ and $\mathcal{L}$ represent resources, blank nodes (anonymous resources), properties and literals respectively. In such dataset, an *entity* $e$ is either a resource or a blank node, that is, $e \in \mathcal{R} \cup \mathcal{B}$. We denote by $D$ the set of entities of the dataset $\mathcal{D}$.   We define a function, denoted $\overline{\phantom{e}}$, which returns the set of properties of an entity: $\overline{e} = \{p \in \mathcal{P} \mid \langle e, p, o \rangle \in D\}$. This function can be extended for a set of entities $E \subseteq D$: $\overline{E} = \bigcup_{e \in E} \overline{e}$. The dataset $D$ is described by the schema $S$, defined as follows.

**Definition 1.** *A schema $S$ describing a dataset $D$ is composed of a set of classes $\{C_1, \ldots, C_n\}$, where each $C_i$ is described by the set of properties $\overline{C_i} = \{p_1^i, \ldots, p_{m_i}^i\}$.*

Consider that over time, new sets of entities are added incrementally to the dataset $D$. The addition of a set of entities denoted $\Delta_D$ to the dataset $D$ may result in $S$ to become incoherent with the new dataset $D \cup \Delta_D$.

To deal with this problem, we make the following assumptions:

1. The dataset $D$ and the newly inserted set of entities $\Delta_D$ can both be massive.

---

[2] https://www.dbpedia.org/
[3] https://github.com/BOUHAMOUM/incremental_sc_dbscan.git

2. The schema $S$ describing the dataset $D$ has been generated using a density-based clustering approach. Among the clustering algorithms, our work focuses on density-based clustering (DBSCAN) [7] which has been used for schema discovery on RDF data and has provided good results [11,3]. We assume in the present work that the schema is produced using this algorithm.
3. The entities of the dataset are compared using the *Jaccard index* which is defined as the size of the intersection of the property sets divided by the size of their union [10]: $\forall e_i, e_j \in D, J(e_i, e_j) = \frac{|\overline{e_i} \cap \overline{e_j}|}{|\overline{e_i} \cup \overline{e_j}|}$

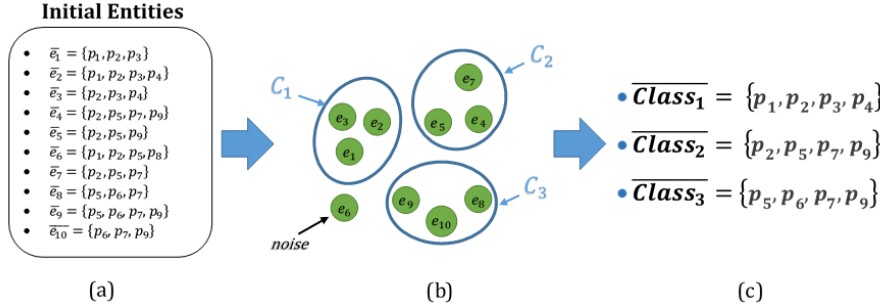

Fig. 1: Example of a Set of Entities and the Corresponding Schema

Figure 1 presents a set of entities (figure 1.a) grouped into three clusters (figure 1.b) using DBSCAN. The similarity threshold $\epsilon$ is set to 0.7 and the density threshold $minPts$ to 2. The resulting clusters represent the classes of the schema (figure 1.c).

In this work, our aim is to update the schema $S$ considering the entities within $\Delta_D$. In order to update this schema, we have to modify the classes impacted by the insertion of the new entities, or create new classes when necessary. The resulting schema after the propagation of updates in the set of existing classes is a descriptive schema which represents the whole dataset, consisting of both the initial dataset $D$ and the set of newly inserted entities $\Delta_D$.

## 3    General Approach

We design in this paper an incremental, distributed, density-based clustering algorithm to extract a schema from large RDF datasets that evolve over time. It allows to keep the schema coherent with the dataset when new entities are added. In order to efficiently manage incrementally growing big datasets, the clustering is restricted to new entities and their neighborhood within the old entities. Clustering the new entities and updating the clusters within their neighborhoods ensures providing the same result as executing DBSCAN on the global data [6].

Our approach is composed of three main steps parallelized and implemented using big data technology. Figure 2 illustrates these different steps.

First, data are split into subsets, called *chunks*, in order to distribute the entities over the different processes (see figure 2.a). The chunks contain entities

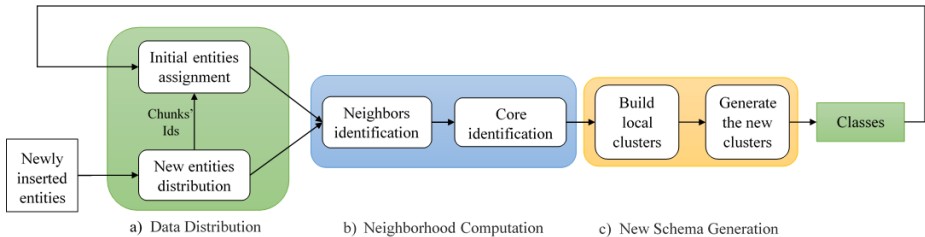

a) Data Distribution        b) Neighborhood Computation        c) New Schema Generation

Fig. 2: Overview of the Incremental Schema Discovery Approach

sharing some properties and which are likely to be similar. The new entities from $\Delta_D$ are distributed, then the identifiers of the created chunks are used for the assignment of old entities from $D$. This way, all the entities that could be similar to new ones, whether in $D$ or $\Delta_D$, are grouped in at least one chunk.

Second, in parallel on each node, the neighborhood for each new entity is computed (see figure 2.b). At the end of this step, entities having dense neighborhoods, called *core entities*, are identified.

Finally, based on the neighborhood of the new entities, the set of clusters is built locally in each chunk. The clusters produced within each chunk are then merged to generate the new clusters that represent the classes of the new schema as illustrated in figure 2.c.

We have implemented our algorithm using Spark [18], a big data technology offering a fast distributed execution of the approach and allowing to manage massive datasets. The following sections detail our proposal.

## 4    Data Distribution Principle for Neighborhood Computation

Computing the neighborhood of the new entities may require a very high number of comparisons. We propose to distribute these new entities according to the distribution principle introduced in [3], where the entities of the dataset are split into different subsets according to their properties. The comparison of entities is performed within each chunk in parallel, thus speeding up the clustering process.

The intuition behind our distribution method is to group entities sharing some properties into chunks to ensure that all the pairs of similar entities will be detected. Indeed, according to the similarity index, two entities are similar if they share a number of properties higher than a given threshold. Thus, entities that could be similar are grouped together in at least one chunk, and will be compared during the computation of their neighborhood. If two entities are not grouped in any of the resulting chunks, this means that they are not similar. This distribution principle allows to skip meaningless comparisons as the similarity between entities in different chunks is not evaluated.

In this section, we first describe how to split the new dataset into chunks, then we show how to assign the initial entities to the created chunks by identifying the ones that could be similar to one of the newly inserted entities.

### 4.1 Distributing New Entities Over Chunks

In our incremental algorithm, we distribute the new entities according to the properties describing them. An entity is distributed according to its properties over several chunks to ensure that it will be compared to all of its neighbors. To optimize the distribution of entities in our approach, we do not consider all the properties of the entities. Thus we limit the duplication of entities in the different chunks and reduce the cost of the comparison process by skipping useless comparisons.

To this aim, the notion of *prefix-filter* is adapted [4]. The intuition behind this notion is that, to be similar, two sets have to share a sufficient number of elements. This number of elements depends on the similarity threshold and on the size of the sets. Moreover, elements must always be chosen in the same order, and thus a total ordering on the elements has to be defined. This result allows to filter candidates considering only their prefix.

From this notion, we define a *dissimilarity threshold* for an entity $e$ as follows:

**Definition 2.** *Let $\epsilon$ be the similarity threshold chosen by the user. The* dissimilarity threshold *for an entity $e$ is the number $dt_\epsilon(e) = |\bar{e}| - \lceil \epsilon \times |\bar{e}| \rceil + 1$.*

This threshold represents the number of properties to consider in order to decide whether this entity could be similar to any other one. It allows to reduce the number of entities to be considered when searching for the neighborhood of a given entity. Note that the *dissimilarity threshold* as defined in our work is based on the Jaccard similarity index. Using another index would require to propose another definition of this threshold based on this index.

As mentioned above, in order to choose the properties for the prefix, we define a total ordering on the properties.

**Definition 3.** *Let $<_{\mathcal{P}}$ be a total order on the properties describing a dataset, $e$ an entity with $\bar{e} = \{p_1, p_2, \ldots, p_n\}$ and $p_i <_{\mathcal{P}} p_{i+1}$ for $1 \leq i < n$. The* comparison set *of $e$ denoted by $cs(e)$ is the set of properties $\{p_1, p_2, \ldots, p_{dt_\epsilon(e)}\}$.*

We will now introduce the definition of a *chunk*.

**Definition 4.** *A* chunk *for a property $p \in \mathcal{P}$ denoted by $[p]$ is a subset of entities having the property $p$ in their comparison set: $[p] = \{e \mid p \in cs(e)\}$.*

Previous results about prefix-filter ensure that by comparing only entities inside chunks, all the comparisons required for the clustering will be performed at least once [4]. The proof of the correctness of this proposition is provided in [3]. For example, if $\epsilon = 0.7$, the entity $e'_1$ described by $\bar{e'_1} = \{p_1, p_5, p_8\}$ is assigned to the chunk $[p_1]$ since $dt(e'_1) = 1$ and $cs(e'_1) = \{p_1\}$, and $e'_2$ described by $\bar{e'_2} = \{p_1, p_3, p_5, p_8\}$ is assigned to $[p_1]$ and $[p_3]$ since $dt(e'_2) = 2$ and $cs(e'_2) = \{p_1, p_3\}$. These two entities are similar, they are grouped and compared in $[p_1]$.

Algorithm 1 describes the distribution of the new entities over the chunks. It requires as input the list of new entities and the similarity threshold $\epsilon$. The distribution of entities is performed in parallel and defines for each entity the

---

**Algorithm 1** Distributing new entities

---

**Require:** the new dataset $\Delta_D$, the similarity threshold $\epsilon$
 1: **for all** entity $e'$ in $\Delta_D$ **do in parallel**
 2:     **for all** property $p \in cs(e')$ **do**
 3:         $[p] = [p] \cup \{e'\}$
 4: Merge the chunks generated by the parallel execution for the same properties
 5: **return**  the chunks

---

chunks it is assigned to (line 1-3) resulting in partial chunks, which are then merged to build the final chunks.

Entities of $\Delta_D$ are distributed over chunks. As they can be in the neighborhood of entities of $D$, we need to identify which entities of $D$ have to be added to the generated chunks. This is the focus of the following subsection.

### 4.2   Assigning Initial Entities to Chunks

As previously stated, the clusters that could be updated due to the insertion of new entities are those within the neighborhood of the new entities. Thus, the entities in $D$ that are in the neighborhood of a newly inserted entity have to be identified. To this end, old entities that share common properties with the new ones are distributed over the generated chunks. By initial entities, we mean the entities in the dataset $D$ prior to the addition of $\Delta_D$, the set of new entities.

To distribute the entities in $D$, we first determine which properties have to be considered: for each entity $e \in D$, we compute its comparison set $cs(e)$ to select the properties to be considered in order to determine the chunks it will be assigned to. The entities are assigned to the existing chunks according to their comparison set. Note that no new chunk is created: old entities are only assigned to chunks already created during the distribution of the new entities. An old entity $e$ is assigned to a chunk $[p]$ if $p \in cs(e)$ and $\exists e' \in \Delta_D, e' \in [p]$. For example, suppose that the created chunks are $[p_1]$ and $[p_3]$. The old entity $e_2$ described by $\overline{e_2} = \{p_1, p_2, p_3, p_4\}$ is assigned to the chunk $[p_1]$. Indeed, $cs(e_1) = \{p_1, p_2\}$, however, the chunk $[p_2]$ is not created and $e_1$ is only assigned to $[p_1]$.

The distribution principle used in this paper ensures that each new entity is grouped with all its candidate neighbors in $D \cup \Delta_D$. New entities are compared with all their candidate similar entities in order to define their neighborhood, and then the clusters that should be updated or created are identified.

## 5   Computing the Neighborhood of the New Entities

In order to propagate the insertion of new entities into the existing schema, we need to compute the neighborhood of the new entities considering both the newly added entities and the old ones which have been previously assigned to existing clusters. This section first describes neighborhood computation for each new entity, then presents the identification of core entities in order to build the clusters.

As the chunks contain entities which are likely to be similar, the $\epsilon$-*neighborhood* of a new entity is identified by computing the similarity between this new entity and all the other ones in the same chunk. We evaluate the similarity between two entities $e_i$ and $e_j$ using the *Jaccard index*.

**Definition 5.** *The $\epsilon$-neighborhood of an entity $e'$ is the set of entities similar to $e'$ with a threshold of $\epsilon$: $neighborhood_\epsilon(e') = \{e \in D \cup \Delta_D \mid J(e',e) \geq \epsilon\}$*

We distinguish between three kinds of entities: *core entities* with at least $minPts$ entities in their $\epsilon$-*neighborhood*, *border entities*, that are not core entities but have at least one core entity in their $\epsilon$-*neighborhood*, and *noise entities*, that are not core entities and have no core entity in their $\epsilon$-*neighborhood*. The latters are never assigned to a cluster.

The $\epsilon$-*neighborhood* is computed for each new entity $e'$ in each chunk by comparing $e'$ to all the entities (new or old) within the same chunk. The $\epsilon$-*neighborhood* is calculated in parallel in the different chunks, independently. The computation of the $\epsilon$-*neighborhood* of the old entities is not required since they have already been clustered in previous iterations. However, the neighborhood of an old entity is updated if it is similar to a new entity. Indeed, old entities that were either border or noise entities can become cores or borders, which would result in updating the old clusters.

Since the neighborhood of entities can be distributed over different chunks, the neighbors discovered in each chunk are consolidated, and the list of neighbors for each entity in the whole dataset is built.

This process leads to the identification of the *core entities*, from which the clusters will be initiated; the cores are the entities having a number of neighbors greater or equal to $minPts$. The old border and noise entities that are similar to new ones can become core or border entities; adding new entities to their $\epsilon$-*neighborhood* could make the number of their neighbors higher or equal to $minPts$ and they will therefore become core entities, or they can be neighbors of a new core. As a consequence to such change occurring for an old entity, the clusters existing prior to the insertion of the new entities have to be updated.

Old entities that are not similar to a new one within a chunk are removed since they will not induce any change on the existing clusters and they will not be assigned to any new cluster.

## 6    Generating the New Schema

In order to update the schema, we first modify the clusters locally in the chunks based on the neighborhood of the new entities. This is performed in parallel within each chunk, providing the local clusters, which are then processed in order to determine the ones that have to be merged. Finally, the new schema is generated by propagating the updates on the old clusters.

### 6.1   Updating Clusters in Each Chunk

After adding the set of entities $\Delta_D$, three situations may occur: (i) existing clusters could be updated by adding new elements, (ii) some clusters could be merged and (iii) new clusters could be created from new core entities.

In a density-based clustering algorithm, the clusters are built according to the density-reachability principle, introduced by the DBSCAN algorithm [7].

**Definition 6.** *An entity $e$ is* density-reachable *from an entity $e'$ wrt. $\epsilon$ and minPts if there is a chain of entities $e_1, \ldots, e_z$, $e_1 = e'$, $e_z = e$ such that $e_{i+1}$ is a core entity and $e_i$ is in its $\epsilon$-neighborhood, $\forall i \in \{1, \ldots, z\text{-}1\}$.*

Based on the core entities, the following change operations can be performed:

- If the $\epsilon$-*neighborhood* of a new core $e' \in \Delta_D$ contains an old core entity $e \in D$ which belongs to an old cluster $C$, then the entity $e'$ is assigned to $C$ and $C$ is also expanded with entities that are density-reachable from $e'$.
- If a core entity $e \in D \cup \Delta_D$ has no old core entity in its $\epsilon$-*neighborhood*, then a new cluster is created and the entities that are density-reachable from $e$ are added to this cluster.
- If the $\epsilon$-*neighborhood* of a core entity $e \in D \cup \Delta_D$ contains two of more old core entities, which belong to distinct clusters, then these clusters are merged and the resulting cluster is expanded with the entities that are density-reachable from $e$.
- If an old core entity has a new entity which is not a core within its neighborhood, then the corresponding new entity is absorbed by the cluster containing this old core entity.

Note that the number of cores is lower than the total number of entities within a chunk. Therefore, iterating over the cores instead of all the entities improves the efficiency of the process.

During this stage, we update the clusters in the neighborhood of the new entities according to the rules defined above. These rules are executed in parallel in the different chunks based on the neighborhood of the entities. Updating the clusters in each chunk is performed considering similar entities within this chunk, providing local clusters.

Algorithm 2 describes the update of the set of clusters within each chunk. It iterates over each core entity within the chunks (line 3); these core entities could be new entities or old ones that have a newly inserted entity in their neighborhood. Then, the algorithm identifies the cluster of the current core in order to expand it (line 6-7) or create a new cluster for this core (line 9), and the cluster is expanded by adding the neighbors of the core (line 10). Next, the algorithm identifies among the added neighbors, those which are cores (line 11), and adds their neighbors to the cluster if they do not belong to any other cluster (line 12-13). If the created cluster $C$ contains a core entity that belongs to another cluster $C'$, then these two clusters are merged (line 15-16).

At the end of this stage, clusters are produced in each chunk. The next section describes the process of building the final clustering result.

---

**Algorithm 2** New Local Clusters

---

**Require:** $CH$: the chunks, $Cores$: the new core entities
 1: **for all** $[p] \in CH$ **do in parallel**
 2:     is-visited $= \emptyset$
 3:     **for all** $e \in Cores$ **do**
 4:         **if**  $e \notin$ is-visited  **then**
 5:             is-visited $=$ isVisited $\cup \{e\}$
 6:             **if**  $e.cluster \neq null$  **then**
 7:                 $C = e.cluster$
 8:             **else**
 9:                 Create a new cluster $C = \{e\}$
10:             $C = C \cup neighborhood_{\epsilon}(e)$
11:             **for all** $e' \in C \mid e' \in cores$ and $e' \notin$ is-visited **do**
12:                 **if**  $e'.cluster = null$  **then**
13:                     $C = C \cup \{e'\} \cup neighborhood_{\epsilon}(e')$
14:                 **else**
15:                     $c' = e'.cluster$
16:                     $c = c \cup c'$
17:         local-clusters $=$ local-clusters $\cup C'$
18: **return**  local-clusters

---

## 6.2   Generating the Final Clusters

Due to data distribution, some clusters may span across multiple chunks. First, the clusters updated independently within the chunks could have elements distributed into different chunks. These clusters share some core entities and will therefore be merged. Second, the clustered entities are either new entities or old entities in the neighborhood of new ones. In order to provide the final result, the updates performed on the clusters have to be propagated in the old entities which have not been distributed in the chunks and have not been considered during the clustering.

In this section, we first describe the identification of the clusters that span across several chunks, and the way the corresponding local clusters are merged. This process is executed on one computing node and is not parallelized. Then, we present the generation of the new schema according to the computed clusters.

According to the density-based clustering algorithm, an entity $e$ is assigned to a cluster $C$ if $e$ is density-reachable from a core entity in $C$. If this same entity $e$ is also in another local cluster $C'$, $e$ is also density-reachable from a core entity in $C'$. If $e$ is a core, it represents a bridge between the entities in the clusters $C$ and $C'$, making them density-reachable. The clusters that span across several chunks are therefore identified by finding out the local clusters that share a common core entity within the newly inserted entities. These clusters are merged to produce the final result. For example, the clusters $C_{p_1.1}$ and $C_{p_3.1}$ produced respectively within the chunks $[p_1]$ and $[p_3]$, are merged if they share a common core entity $e'_1$. The border entities assigned to different clusters are randomly assigned to one of these clusters.

After producing the final clusters representing the new entities and their neighborhood, this result is propagated in the old clusters to construct the new schema. The old clusters to consider at this stage are those which have been merged as a consequence of the insertion of a new core in their neighborhood. The entities previously assigned to these old clusters should therefore be re-assigned to the new cluster resulting from the merging.

If two old clusters $C_i$ and $C_j$ are merged to produce a new cluster $C'$, all the elements of these clusters should be assigned to $C'$. However, not all the old entities are distributed over the chunks. We therefore need to change the assignment of old entities which have not been distributed in the chunks and which belong to clusters that have been merged into a new one.

Finally, all the entities that are not assigned to a cluster, are considered as noise.

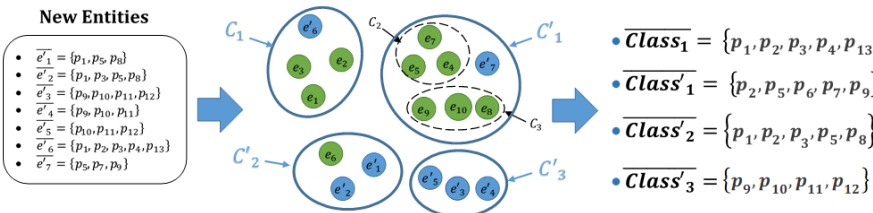

Fig. 3: Updating the Schema after the Insertion of New Entities

Figure 3 presents the updates on the classes introduced in figure 1 following the insertion of a set of new entities. For instance, the set of properties describing $class_1$ has been updated in order to represent the new entity $e'_6$ within the corresponding cluster $C_1$. The classes $Class_2$ and $Class_3$ are merged into $Class'_2$ since the corresponding clusters $C_2$ and $C_3$ have a common core $e'_7$ that is similar to one of their entities, $e_4$ and $e_9$ respectively. Additionally, new classes ($class'_2$ and $class'_3$) are created, representing the newly generated clusters.

This process provides the final clusters, ensuring that they are the same as the ones a sequential DBSCAN algorithm would have generated if executed on the global dataset in one batch.

## 7    Experimental Evaluations

As previously explained, clustering a dataset using our incremental approach provides the same result as clustering the dataset with the DBSCAN algorithm in one batch. This feature of our approach is important since it ensures the good quality of the extracted schema when using DBSCAN for clustering RDF datasets, which has been shown in previous works [11,3].

In this paper, our experiments are therefore focused on the performances of our approach when applied to large evolving datasets. In our experiments,

we evaluate the efficiency of our incremental clustering algorithm compared to the scalable DBSCAN proposed in [3], and derive the speed-up factor when using our incremental approach to reflect the insertion of sets of entities in the clustering result instead of using the scalable DBSCAN algorithm on the dataset composed of the old entities and the newly inserted ones. Both algorithms rely on the Apache Spark 2.0 framework. We have used our implementation of the scalable DBSCAN algorithm, available online[4].

Each time a set of entities $\Delta_D$ is added to the initial dataset $D$, we evaluate the execution time needed by our incremental algorithm to update the clustering result obtained on $D$ so as to reflect the insertion of $\Delta_D$. The execution time of this scenario is compared to the execution time needed by the scalable DBSCAN algorithm in order to cluster the dataset composed of both the initial dataset and the inserted set of entities, i.e. $D \cup \Delta_D$.

First, we have used a synthetic multidimensional dataset of 4 million entities, generated using "IBM Quest Synthetic Data Generator" [5]. In our context, the generator produces the properties of each entity that will be used in our experiments. Second, as the complexity of our incremental approach depends on the number of inserted entities, we have therefore evaluated the incremental algorithm by inserting sets of entities of different sizes. Finally, we illustrate the efficiency of our approach on real datasets. To this end, we apply our approach on 1.2 million entities extracted from DBpedia[6] [1]. All the experiments have been conducted on a cluster running Ubuntu Linux consisting of 5 nodes (1 master and 4 slaves), each one equipped with 30 GB of RAM and a 12-core CPU.

We have first evaluated the scalability of our approach and compared it to the scalable DBSCAN algorithm using several synthetic datasets where we have added datasets of different sizes. Figures 4a, 4b and 4c show both algorithms' runtime as a function of the dataset size. The scalable DBSCAN takes as input the global dataset while the incremental algorithm takes as input the clusters of the previous execution and the newly inserted entities.

The results show that clustering a small dataset is faster using the scalable DBSCAN than using the incremental DBSCAN. This is due to the fact that clustering a small number of entities is very fast and requires a few seconds (22 seconds to clusters 200k entities). Besides, the incremental algorithm executes extra operations such as the assignment of old entities and the union of the result produced by this assignment with the chunks created during the distribution of the new entities, which makes it slower on small datasets compared to the scalable algorithm.

However, when the number of entities is higher, clustering a dataset using the incremental DBSCAN algorithm is faster. This is due to the fact that the clustering is applied on new entities and their neighborhoods, which counterbalances the extra operations, while the scalable DBSCAN algorithm has to build the clusters by computing the neighborhood of all the entities, which is a more

---

[4] https://github.com/BOUHAMOUM/SC-DBSCAN
[5] IBM QSDG: https://sourceforge.net/projects/ibmquestdatagen/
[6] http://downloads.dbpedia.org/3.9/

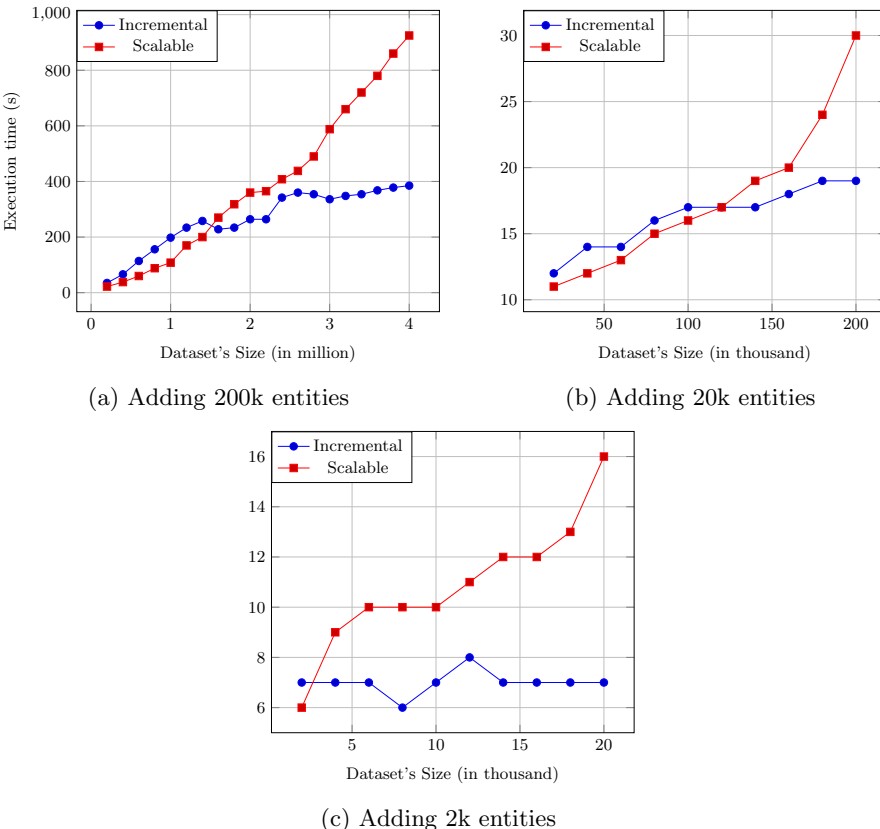

(a) Adding 200k entities          (b) Adding 20k entities

(c) Adding 2k entities

Fig. 4: Incremental vs. Sequential Scalable Algorithm

expensive operation. In addition, the incremental approach produces a lower number of new clusters compared to the scalable algorithm. Thus, when merging the clusters determined within each chunk, a process which is executed in one node, the incremental algorithm has to deal with a lower number of clusters which makes it faster.

We can observe that the bigger the dataset, the larger the gap between the execution time of both algorithms, and the higher the gain achieved by the incremental approach.

Since the complexity of the incremental algorithm is defined by the number of new entities and their neighborhood, we have experimented the insertion of sets of entities $\Delta_D$ of different sizes. The results show that the advantage of the incremental algorithm compared to the scalable DBSCAN is noticed at different levels according to the size of the added set of entities. The smaller the sets of added entities, the faster the clustering using the incremental algorithm. In our experiments, when adding 200k entities at each step, the incremental algorithm becomes faster than the scalable algorithm when the whole dataset reaches the size of 1.6M entities, while when adding 20k at each step, it becomes faster when

the dataset reaches the size of 140k entities (figure 4b). When the size of the inserted datasets is smaller, the gain achieved by the incremental algorithm is more important, as shown in figure 4c after the second insertion. These results are explained by the fact that the incremental algorithm generates the clusters only for the new entities and their neighborhood. It does not take into consideration all the dataset. The smaller the inserted set of entities, the fewer the number of entities which have to be managed by the algorithm, which makes its execution faster.

Finally, we have evaluated the efficiency of our approach on real datasets. Figure 5 illustrates the ability of our incremental algorithm to cluster real datasets, such as DBpedia, a large RDF source from which we have extracted more than 1.2 million entities. Similar to the evaluations on the synthetic datasets, we have added in each insertion to the initial dataset $D$, a set of entities $\Delta_D$ containing 100k entities. Then the execution time of the incremental algorithm is compared to the scalable DBSCAN when executed on the entire dataset.

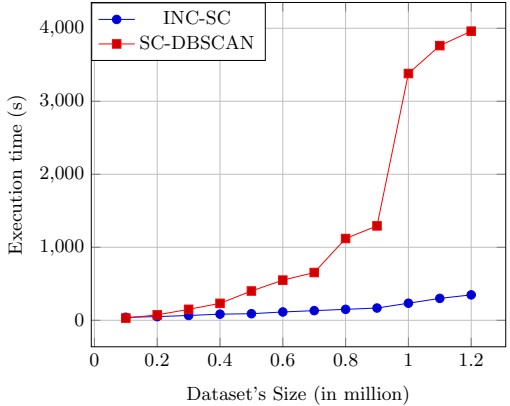

Fig. 5: Clustering a Subset of DBpedia

This evaluation shows that the incremental algorithm overcomes the scalable algorithm in terms of performances. In addition, entities in DBpedia have a high number of properties; some entities have more than 600 properties. As a consequence, the scalable algorithm creates big sized chunks; this has a negative impact on its performances because it reaches the calculation's limit of the cluster when computing the $\epsilon-$neighborhood of the entities, as we notice on the dataset having 1 million entities. However, the incremental algorithm is not impacted by entities having a high number of properties since it manages in each clustering only a limited subset of the dataset and computes the $\epsilon$-*neighborhood* for the new entities only.

## 8   Related Work

Several approaches have been proposed for schema discovery in RDF datasets. Some of them use clustering algorithms to group similar entities in order to form the classes of the schema [5,11]. Other approaches have used frequent pattern mining algorithms to find the most frequent properties describing the schema [17]. However, these approaches have not dealt with scalability issues, and do not scale to process large datasets.

To manage the incrementality issues, the approach presented in [11] proposes a supervised learning step in order to define the type of a new incoming entity, by introducing the concept of fictive entity representing a class, and by comparing a new entity to each fictive entity to determine its type. However, the goal of this approach is to assign an existing type to an instance, and it does not generate new types.

Some approaches have specifically addressed the scalability of schema discovery, providing algorithms capable of managing large datasets, implemented using big data technology such as Hadoop or Spark. However, unlike our approach, these algorithms rely on type declarations to group entities into classes, and then provide a representative schema to help understand the data [2,15]. Such approaches can not be used when these declarations are not provided. To the best of our knowledge, there is no proposal addressing schema discovery for massive RDF datasets without the assumption that type declarations are provided in the dataset.

Our clustering algorithm is inspired by DBSCAN, which is well suited to the requirements of RDF datasets, mainly because it provides clusters of arbitrary shape, which is important in our context where entities of the same type can be described by heterogeneous property sets. However, the main weakness of DBSCAN is its computational complexity which is $\mathcal{O}(n^2)$, where $n$ is the number of entities.

Many works have proposed scalable DBSCAN algorithms by parallelizing their execution, such as [16,12,9], but these approaches are not incremental. Using these algorithms on an evolving dataset would require repeating their execution on the global dataset after each insertion.

Some approaches have proposed an incremental version of DBSCAN. In [6], the neighborhood of an inserted or deleted entity is computed and some rules are proposed in order to update the corresponding clusters. However, this approach processes one entity at a time. In addition, updating the clusters after the insertion of an entity requires its comparison with the entire dataset, which is a costly operation. [13] proposes to enhance the previous approach by limiting the search space during the neighborhood computation. The dataset is split into partitions based on partition centers, and a new entity is assigned to the partition with the closest center. The neighborhood of the new entity is computed within this partition only. However, defining a center in an RDF dataset is not straightforward. In addition, partitioning data based on centers does not ensure that the result is the same as the one of the DBSCAN algorithm, which could decrease the quality of the clustering. RT-DBSCAN [8] proposes to define

the $((minPts\text{-}1) \times \epsilon)$-neighborhood of the new inserted entity and to perform the clustering in this region using DBSCAN. It parallelizes the execution of the approach by dividing the dataset into cells where the incremental algorithm is executed in parallel, then the clusters produced for each cell are merged to build the final clustering result. This algorithm is implemented using Spark streaming. However, this approach is designed for data represented in a 2D space and is not suitable for RDF data.

## 9   Conclusion

In this work, we have addressed the problem of incremental evolution of the schema of large RDF datasets as new entities are inserted.

We have proposed a novel incremental density-based clustering algorithm which scales up the schema discovery process, making it effective for very large RDF datasets. It builds the clusters which group similar entities by updating the existing clusters or creating new ones according to the neighborhood of the newly inserted entities, and ensures that the resulting set of clusters is the same as the one generated using DBSCAN on the global dataset. The clusters produced by our approach represent the classes of the schema, which capture the structure of the entities contained within an RDF dataset. Our proposal has been implemented using Spark, which has enabled the clustering of large RDF datasets. The performed experiments have shown that incrementally extracting a schema from an RDF dataset using our approach outperforms the existing scalable schema discovery approach using scalable DBSCAN when applied on the global dataset, with both synthetic and real data.

In our future works, we will explore the possible ways of enriching the set of classes provided by our approach, by generating the semantic links between these classes as well as providing some semantic annotations. Besides, as some schema-related declarations could be available in the dataset, another possible way of improving our approach is to extend our algorithms in order to exploit partially available schema-related declarations to guide the discovery process, which could improve significantly the quality of the resulting schema.

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
