# OpenReview forum: "Incremental Schema Discovery at Scale for RDF Data"
_eswc-conferences.org/ESWC/2021/Conference/Research_Track — Accept_

### Official Review · AnonReviewer2 · 2020-12-31
**A formal proof of the correctness of the algorithm is missing.**

**Confidence:** 4
**Impact:** 2
**Design And Technical Quality:** 3

**Review:**

*Thanks for the rebuttal. It has addressed some issues I raised. However, I still have concerns about the generalizability of the presented implementation. Therefore, I decided to not change my review score.*

This paper extends [3] by presenting an incremental version of the DBSCAN clustering algorithm for clustering RDF entities based on the Jaccard similarity between their property sets. The algorithm is implemented in a parallel setting. The paper is clearly related to the conference. The paper is easy to follow, but I have concerns about the correctness of the presented algorithm.

**Anonymity:**

Yes, I would like my review to remain anonymous.

**Rating:**

-1: Weak Reject

**Reuse And Availability:**

4: High

**Strong Points:**

(1) The motivation is clear, and the algorithm looks promising.
(2) The experimental design is reasonable.
(3) The paper is easy to read.

**Subreviewer:**

I submitted this review.

**Weak Points:**

(1) Correctness of the algorithm

This is my major concern. The paper mentioned several times that the proposed incremental algorithm and the original DBSCAN algorithm are ensured to provide the same result. I am not sure how this conclusion was drawn. Is it from [6] because [6] is cited at the beginning of Section 3? If not, a formal proof of this claim must be provided to justify the correctness of the algorithm because it is not straightforward.

(2) Generalizability

In Section 2, three assumptions are made. It seems that the proposed technique can only apply to DBSCAN with Jaccard similarity. It would limit the value of this work. I would like to see a discussion about the generalizability of the proposed technique.

(3) Lack empirical comparison

According to the comparison in Section 8, the proposed algorithm is comparable with [6]. I would expect to see such an empirical comparison, while in the current experiments the proposed incremental algorithm was only compared with the original DBSCAN algorithm but not any incremental/parallel implementation.

(4) Minor issues

P7: is parallel --> in
P9: than span --> that
P11: have add --> added

---

> ### Author Rebuttal · Authors · 2021-01-29
>
> We would like to thank the reviewer for the comments and the remarks.
>
> 1. Correctness of the algorithm:
> This is my major concern. The paper mentioned several times that the proposed incremental algorithm and the original DBSCAN algorithm are ensured to provide the same result. I am not sure how this conclusion was drawn. Is it from [6] because [6] is cited at the beginning of Section 3? If not, a formal proof of this claim must be provided to justify the correctness of the algorithm because it is not straightforward.
>
>    - We will add the required explanation in the paper. In [6], it has been proved that updating the clusters within the neighborhood of an inserted entity is sufficient to provide the same result as the original DBSCAN.
> We use this well known result by determining the neighborhood of an inserted entity within both the new and the old entities. The distribution principle ensures that each entity is grouped with all its neighbors.
> Our approach thus provides the same clustering result as DBSCAN.
>
> 2. Generalizability:
> In Section 2, three assumptions are made. It seems that the proposed technique can only apply to DBSCAN with Jaccard similarity. It would limit the value of this work. I would like to see a discussion about the generalizability of the proposed technique.
>
>    - In our approach, we assume that the existing schema has been extracted using DBSCAN. The generalization to other clustering approaches is possible but would required adapting the distribution principle.
> As for the similarity index, other indexes could be used, and this would required the redefinition of the dissimilarity threshold, as the one defined in this work is related to Jaccard.
>
> 3. Lack empirical comparison:
> According to the comparison in Section 8, the proposed algorithm is comparable with [6]. I would expect to see such an empirical comparison, while in the current experiments the proposed incremental algorithm was only compared with the original DBSCAN algorithm but not any incremental/parallel implementation.
>
>    - The incremental algorithm has not been compared to the sequential incremental DBSCAN [6], but to the one proposed in [3] which is an scalable schema discovery approach based on DBSCAN.

---

### Official Review · AnonReviewer4 · 2021-01-14
**Incremental Schema Discovery for RDF Data at Scale**

**Rating:** 1
**Confidence:** 4
**Impact:** 4
**Design And Technical Quality:** 2

**Review:**

Incremental Schema Discovery for RDF Data at Scale

Redouane Bouhamoum, Zoubida Kedad, and Stéphane Lopes

The paper addresses the schema discovery problem in linked data. It extends a previous paper of the same authors with an incremental schema discovery algorithm. The main motivation is to be able to follow the evolution of datasets without re-discovering schema from scratch at each update.
The incremental discovery algorithm ensures providing the same result as if the clustering algorithm has been executed on the whole dataset. Finally, The incremental discovery algorithm can be parallelized to scale on a large dataset.

Strong points
* The motivations of the paper are clear and concern a well-known issue of the semantic web
* The experimental section demonstrates that the incremental algorithm outperforms the previous one.

Weak points
* The paper focuses on the extension of previous work without describing exactly the properties ensured, and the usage of such schema discovery. In its current form, the paper is not self-contained (see my detailed comments).
* The related work section looks incomplete to me.
* Some parts of the proposal remain confusing. Just by reading the paper, I don't think I’m able to reproduce the system.




Section 2 describes the problem statement. Definitions look clear to me and the example is easy to follow. However, the considered schema discovery algorithm is density-based clustering [11,3] and the authors wrote: “provide *good* results”. Good for what? What is the definition of “good”?. Is it better than characteristic sets? If yes, why? Is it better than schemex? If yes, why?
From Figure 1, e6 is considered as noise and does not appear in extracted class. Maybe it is good for building an approximate summary, but if I want to compute source selection with such a schema, I can miss a source. So it is really important for me to know what [11,3] is good for… More generally, as the paper is a kind of extension of [11], [11] has to be precisely described especially on its properties. I regret that section 2 does not end with a clear description of the problem statement itself.


Section 3 describes the general approach and is easy to follow. There are three steps in the algorithm, so a section will describe each step. It is not clear to me what the approach remembers between two ingestion of delta. Is it only the definition of classes or some extra information have to be preserved??




Section 4 describes how the distribution of new entities over chunks. The overall objective here is to partition entities based on the concept of prefix-filter to avoid useless comparisons. Definition 2 uses a kind of bracket I don't know. Can it be more precise? The computations just under definition 4 are not crystal clear to me. How dt(e’1)=1? (maybe impacted by brackets of def 2).


Section 4.2 is quite confusing. The “clusters” are the class identified in a previous step? I’m confused with “old entities”. From figure 2, the input of the initial entities assignment is a list of classes and a set of chunks, so what are the “old entities”? Do the old entities need to be stored somewhere, or just the classes have to be stored between two ingestions ??


Section 5 requires notion coming from density-based clustering (core entities, border entities, etc). This has not been introduced before. Section 5 is not illustrated with an example and remains confusing.


Section 6 describes how to generate the new schema. It mainly describes algorithm 2 without explaining the idea behind the algorithm (maybe present in 6.2). In the introduction, it is written, “The incremental discovery algorithm ensures  providing the same result as if the clustering algorithm has been executed on the whole dataset.” However, I don’t see evidence of that in the paper. I think it is nearly impossible to do that if we cannot compare the original algorithm with the incremental one.


Section 7 describes the experimental evaluations. “...since it preserves the good quality of the extracted schema”. Again, what is the meaning of good? Nevertheless, the experimentation demonstrates that incremental schema discovery outperforms the non-incremental one as expected.


Section 8 presents related works. I’m surprised to not see characteristic sets, but maybe I'm wrong. Anyway, I would like to know why it is not considered. I also wonder why Schemex is not cited.

# After Rebuttal

"We will improve the paper by discussing the usage of schema discovery"
* it would have been nice to give me at least one of them. You wrote, "our goal is to infer the classes of a dataset by grouping its instances based on their structural similarity". So at the end, we will obtain an approximation of the schema, I just wonder
which applications can be written with approximated schema?

"We can also find similarities with other streams of work"
* I agree with the answer, but if the usage of such techniques is the same as summarization, then related works have to be updated.

"We have considered these types as the ground truth and we have compared the generated classes to the ground truth"
* ok.

"can not be compared to Characteristic sets..."
* I agree that CS targets cardinality estimation, but CS "characterizes" an entity by its emitting edges and then can be seen also a good schema-based summary of a dataset.

Anyway, I like the idea that mining algorithms have to scale *and* support incremental changes to be useful.
This is what we have with this paper.


**Anonymity:**

No, I would like my review to be deanonymized.

**Reuse And Availability:**

4: High

**Strong Points:**

* The motivations of the paper are clear and concern a well-known issue of the semantic web
* The experimental section demonstrates that the incremental algorithm outperforms the previous one.


**Subreviewer:**

I submitted this review.

**Weak Points:**

* The paper focuses on the extension of a previous work without describing exactly the properties ensured, and the usage of such schema discovery. In its current form, the paper is not self-contained (see my detailed comments).
* The related work section looks incomplete to me.
* Some parts of the proposal remain confusing. Just by reading the paper, I don't think i’m able to reproduce the system.

---

> ### Author Rebuttal · Authors · 2021-01-29
>
> We would like to thank the reviewer for the remarks and comments.
>
> I. Weak points
> - Indeed, the paper addresses  incremental issues based on the work in [3]. In our view this is a problem in its own rights: without taking the evolution of the dataset into account, the defined schema will soon become obsolete and useless. We will improve the paper by discussing the usage of schema discovery.
> - We have inserted to the best of our knowledge the existing schema discovery approaches, and discussed issues of scalability and incrementality. We can also find similarities with other streams of work such as summarization or annotation for example, but discussing all these would require more space.
>
> II. Reviews
> 1. Section 2 ... authors wrote “provide good results”
>  - In order to evaluate the quality, we have considered a data set containing typed entities. We have considered these types as the ground truth and we have compared the generated classes to the ground truth. The quality metrics (precision and recall) have shown that the approach was able to accurately predict the classes [3].
> In our view, our approach, which extracts the hidden structure of an RDF dataset, can not be compared to Characteristic sets, which is a cardinality estimation method for RDF star joins.
> - Schemex creates an index on distributed RDF datasets in order to accelerate query execution. While our approach discovers the implicit schema in a dataset by grouping similar entities, Schemex groups entities having exactly the same property sets or having the same  $rdf:type$ declaration, which does not lead to classes including entities which are similar.
> 2. From Figure 1, ...
>  - The schema is descriptive and could be used to assess most of the query answering capabilities of a dataset. But of course, it does not encompass all the information. The enrichment of the resulting schema would be an interesting lead for future works.
> 3. More generally, ..., [11] has to be precisely described especially on its properties
>  - The paper is not an extension of [11], but it uses the same well known clustering algorithm and adapts it. The paper is an extension of [3], it focuses on incrementality issues, while [3] is unable to manage dataset evolutions.
> 4. I regret that section 2 does not end with a clear description of the problem statement itself
>  - we will improve the paper by describing precisely the problem of incrementally updating the schema extracted from an RDF data source in order to keep it coherent with the addition of new entities.
> 5. Section 3 ...
>  - The approach remembers the classes and the neighborhood of each entity.
> The list of neighbors is required in order to ensure the same clustering result as if it was performed on the whole dataset. This is because, when the neighborhood of an old entity changes, the clusters are impacted.
> 6. Section 4 ...
>  - In order to optimize the distribution of entities, we do not consider all the properties. The prefix filter assumes that if the properties of two entities differ to a certain extent, these entities cannot be similar.
> The dissimilarity threshold (ds) expresses the number of properties to consider for each entity $e$, in order to state that any other entity with $ds(e)$ different properties cannot be similar to $e$.
> 7. The computations .. How dt(e’1)=1?
>  - The $dt(e’_1)$ for the entity $e'_1$,  such as $\overline{e'_1}=${$p_1,p_5,p_8$} is computed as follows: $dt(e’_1)=|\overline{e'_1}|-(|\overline{e'_1} |*\epsilon)|+1=3-|3 * 0,7|+1=1$
> 8. Section 4.2 ...
>  - Indeed, the old entities are the existing entities grouped into clusters prior to the insertion of the new entities.
> For each old entity, both the cluster it belongs to and the list of its neighbors are stored.
> 9. Section 5 requires notion coming from density based clustering ...
>  - We will improve the definition of these notions.
> 10. Section 5 is not illustrated with an example and remains confusing.
>  - Thank you for the suggestion, we will add an example.
> 11. Section 6 ...
>  - We will improve this section to answer this comment. A well known result [6] proves that updating the neighborhood of an inserted entity is sufficient to provide the same result as DBSCAN. As our approach determines the neighborhood of an inserted entity within both the new and the old entities, it therefore provides the same clusters as DBSCAN.
> 12. Section 7 ...
>  - We will update the paper to clarify the notion of quality, as explained above in I.1.
>
> 13 - Section 8 ...
>  - There are indeed other categories of approaches for discovering the hidden structure of a dataset. Schemex is definitely one of them. In our view, it falls into the category of approaches which extract exact patterns from a dataset.  Due to space limitation, we have chosen to compare our approach to the works which goal is to infer the classes of a dataset by grouping its instances based on their structural similarity. It would definitely be helpful to add this explanation in the paper and we will do.

---

### Official Review · AnonReviewer1 · 2021-01-15
**Clustering vs claimed schema discovery**

**Confidence:** 5
**Impact:** 2
**Design And Technical Quality:** 2

**Review:**

AFTER REBUTTAL
I would like to thank the authors for their response. I'm still concerned about the paper's following issues: i. the schema definition ignores the original data's graph structure, ii. inadequate justification for using density-based clustering, iii. and lack of experiments that show the usefulness of the new clusters.

The paper presents a method to learn schema from massive Knowledge Graphs incrementally. Although the idea of identifying new classes from data to reflect the evolution of data is an interesting idea, the proposed method has some major issues. The definition of the schema is too simplistic and arbitrary. By defining schema in this simplistic way, the multi-relational data is stripped off from its network connectivity. In this way, a KG is presented as a set of entities where each entity has a bunch of predicates as its features. This translation is not lossless and costs the massive information encoded in the original data's graph structure. The work is more focused on clustering entities, instead of learning schema from RDF data. It is not clear how any feature-vector clustering method cannot solve the defined problem (after extracting each entity's property set).
In the experiments, the runtime of the proposed method is compared with the baselines. The evaluation of learned classes based on their informative value is missing. This evaluation could be done via a downstream task; for example, the authors could show by proposing new classes the link prediction task could be done with higher accuracy.

**Anonymity:**

Yes, I would like my review to remain anonymous.

**Rating:**

-2: Reject

**Reuse And Availability:**

3: Medium

**Strong Points:**

Novel approach of identifying classes from data

**Subreviewer:**

I delegated this review to a subreviewer.

**Weak Points:**

Simplistic assumptions about schema
Results are clusters rather than learning schema
Weak evaluation

---

> ### Author Rebuttal · Authors · 2021-01-29
>
> We would like to thank the reviewer for the remarks and comments.
>
> 1. The definition of the schema is too simplistic and arbitrary. By defining schema in this simplistic way, the multi-relational data is stripped off from its network connectivity. In this way, a KG is presented as a set of entities where each entity has a bunch of predicates as its features. This translation is not lossless and costs the massive information encoded in the original data's graph structure.
>    - This work focuses on incrementality issues for class discovery in a big data setting, hence the proposed definition. In future works, we could indeed consider a more complex definition of schema, by including semantic relationships between classes for example.
>
> 2. The work is more focused on clustering entities, instead of learning schema from RDF data. It is not clear how any feature-vector clustering method cannot solve the defined problem (after extracting each entity's property set).
>    - Indeed, other approaches could be used for schema discovery. In our view, density-based clustering is well suited to entities in an RDF dataset as they are irregular. Even if they should be assigned the same type, they might have different property sets. Density-reachability allows to generate clusters of arbitrary shape. We will add an explanation in the paper.
>
> 3. The evaluation of learned classes based on their informative value is missing. This evaluation could be done via a downstream task; for example, the authors could show by proposing new classes the link prediction task could be done with higher accuracy.
>    -  Assessing the learned classes considering a use case such as data interlinking would be very interesting indeed.
> As this work focuses on incrementality issues, we had not the space required for an in-depth evaluation of the quality of the classes. We have provided some of these evaluations in previous work [3].

---

### Official Review · AnonReviewer5 · 2021-01-17
**Incremental Schema Discovery for RDF Data at Scale**

**Rating:** 1
**Confidence:** 4
**Impact:** 3
**Design And Technical Quality:** 3

**Review:**

The paper proposes a novel approach towards incremental schema evolution of large-scale RDF datasets. Its approach uses an incremental density-based clustering algorithm that scales up to massive RDF datasets. It does that by grouping similar entities while updating the existing clusters or creating new ones according to the closest entities of the newly inserted entities. The results are comparable with the original approach DBSCAN on the same dataset. It shows empirically that the proposed approach is scalable, and have shown that incrementally extracting a schema from an RDF dataset outperform the original (DBSCAN) schema discovery approach when applied on an RDF graph. These experiments were conducted on a commodity cluster using real and synthetic datasets.

Positive and negative aspects:

- (+) the paper is well written and organized
- (+) it gives a comprehensive analysis and evaluation
- (+) the motivation is well justified
- (+) it compares with other well-known data partitioning techniques
- (+) the scalability is well supported
- (+) the source code and instructions for reproducing the results are available online
- (-) it lacks on comparison with other scalable/distributed approaches but rather focuses on the effect of the proposed approach as an addition to the original approach DBSCAN.

Overall, the paper is well-written and structured. The motivation for having an incremental density-based clustering approach while addressing the incremental evolution of the schema discovery of large RDF datasets is well placed.
However, in its present form, the paper suffers from a number of minor issues, which are outlined below.
- Wouldn’t it be better to use a different naming when distinguishing between the original DBSCAN (in the paper you refer it as a scalable algorithm) and your approach? I mean, both approaches are scalable, right? But using different approaches on how the clustering has been performed i.e. in your case using the so-called incremental approach. I will recommend that you chose a different name and be consistent throughout the paper.
- I also have a question about the way experiments are conducted. Is the “global” dataset (I wouldn’t call it a global dataset either) re-loaded whenever new entities are added. In real-life scenario i.e. extracting new information from Wikipedia towards DBpedia, we will need to reload the whole DBpedia once more (on the paper you are referring to this a global dataset) as the process is batch processing and therefore you will have to load the dataset from the beginning and then attach new entities (information) in order to be clustered. I would like to see how the approach behaves w.r.t to performance when this scenario is used. Or I may overlook something.

- I do not see a comparison with e.g. DBpedia Information Extraction Framework which would also lead us to a better understanding of the completeness of the approach. The accuracy of the schema identification would be as compared with the DBpedia Extraction Framework?

==== Minor comments ====

Section 2. “An RDF dataset is a set of RDF(S)/OWL triples” – I will remove

**Anonymity:**

Yes, I would like my review to remain anonymous.

**Reuse And Availability:**

3: Medium

**Subreviewer:**

I submitted this review.

---

> ### Author Rebuttal · Authors · 2021-01-29
>
> We would like to thank the reviewer for the remarks and comments.
>
> 1. The results are comparable with the original approach DBSCAN on the same dataset.
>   - The approach was compared to a scalable schema discovery approach based on a scalable DBSCAN [3], not with the original DBSCAN [7].
>
> 2. Wouldn't it be better to use a different naming when distinguishing between the original DBSCAN (in the paper you refer it as a scalable algorithm) and your approach? I mean, both approaches are scalable, right?
>   - The original DBSCAN [7] applied to schema discorery is not scalable. The scalable algorithm we refer to is the one in [3], which has been designed to ensure sclalability.
> We will check and modify the paper throughout to make sure there is no confusion.
>
> 3. But using different approaches on how the clustering has been performed i.e. in your case using the so-called incremental approach. I will recommend that you chose a different name and be consistent throughout the paper.
>   - Indeed, we will modify accordingly.
>
> 4. I also have a question about the way experiments are conducted. Is the “global” dataset (I wouldn’t call it a global dataset either) re-loaded whenever new entities are added. In real-life scenario i.e. extracting new information from Wikipedia towards DBpedia, we will need to reload the whole DBpedia once more (on the paper you are referring to this a global dataset) as the process is batch processing and therefore you will have to load the dataset from the beginning and then attach new entities (information) in order to be
> clustered. I would like to see how the approach behaves w.r.t to performance when this scenario is used. Or I may overlook something.
>   - The term global refers to the union of the existing dataset and the newly inserted set of entities. Our algorithm is not performed on this dataset, but on the union of the set of inserted entities and the set existing entities which are likely to be similar to the new ones.
>
> 5. I do not see a comparison with e.g. DBpedia Information Extraction Framework which would also lead us to a better understanding of the completeness of the approach. The accuracy of the schema identification would be as compared with the DBpedia Extraction Framework?
>   - In our view, the two processes are different: while DBpedia Information Extraction focuses on URI and triple extraction from Wikipedia infoboxes, our problem is rather focused on inferring a missing schema based on the structure of the entities (i.e. their property sets).

---

### Official Review · AnonReviewer3 · 2021-01-17
**A nice incremental schema discovery approach for RDF datasets using density based clustering algorithm**

**Rating:** 2
**Confidence:** 4
**Impact:** 4
**Design And Technical Quality:** 3

**Review:**

Post rebuttal

I would like to thank the authors for their response. Please add the points from the rebuttal, especially the one on correctness in the final manuscript.

------------------------------------

Authors propose an incremental density based clustering algorithm in order to discover schema for RDF datasets that change over time. The changes are propagated to different clusters that correspond to the classes of the schema. The proposed algorithm has been implemented using Apache Spark to make it scalable. In the evaluation, synthetic dataset and a real-world dataset (DBpedia) has been used and the results are encouraging.

The incremental approach discussed in this submission could be of value in practice since the RDF datasets keep getting updated and there will be a need to keep the schema consistent. Added to that, the algorithm is also scalable. Apart from that, I think the approach has been described quite sufficiently in general. It would have helped if an example from a real-world dataset (DBpedia) has been used to discuss Sections 4, 5 and 6.

It would have been good if there is a discussion on the correctness or the appropriateness of the clusters (classes). It was mentioned that the results would be similar to that of DBSCAN, but at least a brief discussion of this aspect would be good, especially because it is claimed that the authors' approach generates fewer clusters compared to the existing work. Apart from that, it would be interesting to see how this algorithm performs on at least one more real-world dataset such as Wikidata.


Other questions/comments

1) How are the new entities obtained for the DBpedia dataset?
2) On page 3, there was an assumption made about the schema generation. Why is this assumption required? Can't we assume that the schema exists and not care of the mechanism behind its generation? For example, it might have been built manually.
3) On page 3, minPts has been used but not defined.
4) Page 5, definition 3, a total order on the properties/predicates has been assumed. But it is not clear what is the (binary) relation here among the items of the set (properties).
5) How do you assign the labels (class names) for the clustered groups?
6) Page 2, Section 2, function name is missing in the first paragraph of this section.
7) Page 7, what are "cores or borders"?
8) Page 7, Section 6, second line, it should "in" instead of "is" (in parallel)
9) Page 9 and 13, it should be "Similar" instead of "Similarly".
10) Page 11, the phrase "... number of entities is more important ..." needs to be rephrased.
11) Page 14, the phrase "... a big data technology such as ..." should be rephrased to "... using big data technologies such as Hadoop and Spark ...".
12) Page 14, it should be "update" instead of "updates" and the phrase "... when a set of several entities is inserted." should be "... when several entities are inserted."
13) Page 15, it should be "cell" instead of "cells". In the Conclusion second line, "a" is not required.

**Anonymity:**

No, I would like my review to be deanonymized.

**Reuse And Availability:**

4: High

**Strong Points:**

1) A scalable approach that seems to be quite useful in practice.
2) In general, the work has been presented well.

**Subreviewer:**

I submitted this review.

**Weak Points:**

1) A description of how good the clusters (classes) are is missing.
2) Evaluating the incremental algorithm on at least one more real-world dataset would have been good.

---

> ### Author Rebuttal · Authors · 2021-01-29
>
> We would like to thank the reviewers for the remarks and comments.
> 1. It would have helped if an example from a real-world dataset (DBpedia) has been used to discuss Sections 4, 5 and 6.
>    - Indeed this would have helped, we could not do this for space limitation only.
>
> 2. It would have been good if there is a discussion on the correctness or the appropriateness of the clusters (classes).
>    - As our focus is on the scalability of the incremental algorithm, we did not include a correctness discussion, which we had previously addressed in [3]. We have included a reference in the present paper.
>
> 3. It was mentioned that the results would be similar to that of DBSCAN, but at least a brief discussion of this aspect would be good, especially because it is claimed that the authors' approach generates fewer clusters compared to the existing work.
>    - In [6], it is proven that updating the neighborhood of an inserted entity is sufficient to provide the same result as the original DBSCAN.
> In our approach, the distribution of both old and new entities ensures grouping each entity with all its neighbors, thus, providing the same clustering result as DBSCAN.
> It is right that during the clustering phase, less clusters are generated since our approach is not applied to the whole datase, but only to the new entities and the old ones in their neighborhood.
>
> Other questions/comments:
>
> 1. How are the new entities obtained for the DBpedia dataset?
>    - The entities are extracted using the pattern extraction approach defined in (*) which provides all the structures (combination of properties) existing among the entities in an RDF dataset.
>
>    (*) Redouane Bouhamoum, Kenza Kenza Kellou-Menouer, Stephane Lopes, andZoubida  Kedad.    Scaling  up  schema  discovery  approaches.    InProceed-ing  of  the  34th  International  Conference  on  Data  Engineering  Workshops(ICDEW), pages 84–89. IEEE, 2018
>
> 2. On page 3, there was an assumption made about the schema generation. Why is this assumption required?
> Can't we assume that the schema exists and not care of the mechanism behind its generation? For
> example, it might have been built manually.
>    - The approach could be adapted by considering other assumptions for the generation of the initial schema, but the incremental evolution of the classes/clusters would have to be redefined to ensure the quality of the resulting schema. The one proposed in this paper is specific to density-based clustering.
>
> 3. On page 3, $minPts$ has been used but not defined.
>    - represents the number of neighbors, we will add the definition.
>
> 4. Page 5, definition 3, a total order on the properties/predicates has been assumed. But it is not clear what is the (binary) relation here among the items of the set (properties).
>    - The total order ensures that similar entities are grouped in the same chunks and compared later. It can be chosen arbitrarily but some choices can improve performances.
> In our evaluations, we used an order based on the selectivity of the properties.
>
> 5. How do you assign the labels (class names) for the clustered groups?
>    - This is a very interesting problem that we did not addressed yet. It will be surely in future works.
>
> 6. Page 2, Section 2, function name is missing in the first paragraph of this section.
>    - The function is represented by the "bar" on top: the annotation $\overline{e}$ represent the properties of the entity $e$.
>
> 7. Page 7, what are "cores or borders"?
>    - A core is an entity having at least $minPts$ neighbors, and a border is an entity within the neighborhood of a core having less than $minPts$ neighbors.
> We have provided these definitions are provided in section 5.

---

### Decision · Program_Chairs · 2021-02-23

**Decision:**

Accept

**Comment:**

The reviewers agreed that the paper is well written an d provides a comprehensive analysis and evaluation. The  idea that mining algorithms have to scale and support incremental changes to be useful is interesting.
The paper presents a nice incremental schema discovery approach for RDF using a density algorithm.
Based on the evaluations of  the reviewers, this paper is recommended for acceptance.

In preparation for the camera-ready, we kindly ask the authors to address the comments raised by the reviewers and include the requested clarification :

1. Check all editing tips.
2. Please add the reference to the proof of the correctness of the algorithm
3. Clarify what is  minPts
4. Add discussion about the generality of the proposed technique